# Impact of variation in functions and delivery on the effectiveness of behavioural and mood management interventions for smoking cessation in people with depression: protocol for a systematic review and meta-analysis

Gemma Taylor,[1,2] Paul Aveyard,[3] Regina Van der Meer,[4,5] Daniel Toze,[1,2] Bobby Stuijfzand,[6] David Kessler,[7] Marcus Munafò[1,2]

For numbered affiliations see end of article.

**Correspondence to**
Dr Gemma Taylor;
gemma.taylor@bristol.ac.uk

## ABSTRACT

**Introduction** Tobacco is the world's leading preventable cause of disease and death. People with depression are twice as likely to smoke and are less responsive to standard tobacco treatments as compared with the general population. A Cochrane systematic review of randomised controlled trials of smoking cessation treatment for smokers with current or historical depression found that adding mood management to usual smoking treatment improved quit rates. However, the review did not examine if variation in intervention delivery or intervention functions impacted on treatment effectiveness. With the aim of providing information to develop tailored approaches to treating smoking for people with current depression, we will add-on to the Cochrane review in three ways: (1) use the Template for Intervention Description and Replication checklist to determine if variations in mood management delivery have impact on intervention effectiveness, (2) use the Taxonomy of Behaviour Change techniques for smoking cessation to examine which behaviour change functions are most effective for smoking cessation in people with current depression and (3) examine the difference in change in depression scores between intervention and control arms.

**Methods and analysis** We will include randomised controlled trials of smokers with current depression as identified by a previous Cochrane review and the in-progress update of this Cochrane review. We will use meta-regression to examine (1) if variations in delivery of mood management impact on smoking cessation intervention effectiveness, (2) determine which behaviour change functions are most effective for smoking cessation and (3) use meta-analysis of the difference in change in depression scores between treatment arms from baseline to follow-up to determine if offering smoking cessation treatment causes psychological harm.

**Ethics and dissemination** Ethical approval is not required for this study. We will disseminate the findings of this work at national and international conferences, and to relevant patient panels.

**PROSPERO registration number** CRD42017070741.

### Strengths and limitations of this study

► We will examine the impact of variation in intervention delivery and intervention functions on treatment effectiveness using peer-reviewed checklists: The Behaviour Change Taxonomy and Template for Intervention Description and Replication.
► We will obtain an estimate of the effect of stopping smoking on depression symptoms.
► The study design may suffer from low power and/or publication bias.

## BACKGROUND

Tobacco is the world's leading preventable cause of disease and death.[1] In the UK and in other developed nations, smoking prevalence has declined substantially in the general population, but has remained largely unchanged in those with mental health problems resulting in an excess burden of smoking-related mortality in this group.[2 3] People with depression are twice as likely to smoke[4 5] and are less responsive to standard tobacco treatments than are the general population[6 7] leading to urgent calls for targeted smoking interventions.[8]

The Cochrane group conducted a systematic review and meta-analysis of smoking cessation interventions for smokers with past or present depression. The review included pharmacological and behavioural interventions to aid cessation and found that adding mood management to a usual smoking treatment (eg, nicotine replacement therapy, telephone counselling and self-help website) moderately increased smoking cessation rates in people with current depression compared with usual smoking treatment

alone, reporting a risk ratio (RR) of 1.47 (95% CI 1.13 to 1.92).[9] The review highlighted the importance of adding psychological techniques to handle depressive symptoms in standard smoking treatments for people with depression. However, in the meta-analysis, there was variation between the included studies' direction of effect and it is possible that this variation may be in part related to differences in intervention delivery or intervention functions, for example. Further investigation into these potential modifiers will provide useful information for development of smoking cessation interventions for people with current depression.[9]

In addition, the review did not examine the impact of behavioural or psychological smoking cessation interventions on depression symptoms. This is an important question as many clinicians believe that smoking may offer mental health benefits, or that their patients' mental health may deteriorate on cessation.[10] However, there are data from meta-analyses of cohort studies indicating that quitting smoking may improve depression,[11] but due to common pitfalls of observational data one cannot be sure that this is a causal association. If treating smoking is found to not worsen depression, then these data can be used to assure clinicians that they are not causing psychological harm by helping their patients to quit smoking.

In our review, we aim to add-on to the 2013 Cochrane review[9] in three ways. We will:

1. use the Template for Intervention Description and Replication (TIDieR) checklist[12] to determine if variations in mood management delivery impact on intervention effectiveness in people with depression;
2. use the Taxonomy of Behaviour Change (BCT) techniques for smoking cessation[13] to examine which behaviour change functions are most effective for smoking cessation in people with current depression;
3. examine the difference in change in depression scores between intervention and control arms in people with current depression.

## METHODS

The study protocol has been registered in advance on the International Prospective Register of Systematic Reviews ((PROSPERO); ID: CRD42017070741; http://www.crd.york.ac.uk/PROSPERO/). All methods and study reporting will adhere to guidance described within the Cochrane Handbook for Systematic Reviews and Meta-analyses of Randomised Controlled Trials.[14]

### Search strategy

We will include relevant studies identified by a previously conducted Cochrane review of smoking cessation interventions for people with depression, and from the Cochrane review update due to commence this year.[9] Studies have been identified from the Cochrane Central Register of Controlled trials (CENTRAL), MEDLINE, EMBASE and PsycINFO using search terms related to 'depression', and 'tobacco' or 'smoking' as

recommended by the Tobacco Addiction Group and the Cochrane Depression, Anxiety and Neurosis Group. See the Tobacco Addiction Group Module in The Cochrane Library for full search strategies and the list of other resources searched.[9] This search strategy will be updated for additional relevant studies published from 2013. RVM is the lead author for the Cochrane review published in 2013, and will lead on the Cochrane update of this review. To avoid duplicating efforts across teams and given the high reliance of Cochrane methods, RVM will share the eligible studies prior to data extraction of the Cochrane update. We predict that this will take place in early 2018.

### Inclusion criteria

Inclusion criteria are based on those outlined in the 2013 Cochrane review.[9]

► study design: randomised controlled trials only;
► participants: daily smokers with current depression, any definition of depression, no restrictions by physical or mental comorbidities;
► intervention: any smoking cessation intervention;
► intervention delivery: self-help, individual, group, internet;
► control: any (eg, including self-help, no treatment, etc);
► outcome: any ascertainment of smoking cessation;
► follow-up: follow-up at a minimum of 6 months from the quit date.

### Outcomes

► Smoking status at final follow-up (the same as the 2013 Cochrane review[9]).
► Change in depression scores from baseline to final follow-up (not reported in the 2013 Cochrane review[9]).

### Data extraction

We will use the following data as reported in the 2013 Cochrane review[9]:

► Trial methods: study design, setting, country, randomisation methods.
► Participants: number of participants per intervention group, definition of depression, type of smoker, comorbid conditions, age, sex, ethnicity, level of education, nicotine dependence, mean/median number of cigarettes per day, depression type and severity.
► Outcomes: smoking cessation status, biochemical validation, depression scores, length of follow-up.
► Measures of treatment effect, smoking cessation (study aims 1 and 2): We will use the following outcome data as reported in the Cochrane review. The number of participants randomised to the intervention and control groups, and the number of participants who quit smoking in the intervention and control groups.[14]

We will extract the following additional data not reported in the 2013 Cochrane review[9]:

**Table 1** Modified version of Template for Intervention Description and Replication (TIDieR) checklist[12] for use in meta-regression analysis*

| Item | Categories |
|---|---|
| Materials for mood management (ie, physical or informational materials used) | Paper-based information, website, homework, diary, audio information, etc |
| Procedures for mood management: activities, procedures or activities used in the intervention to support activities | Relaxation techniques, mood monitoring, etc |
| Did the participant see the same intervention provider for all mood management sessions? | Yes, no |
| Mood management provider | Nurse, psychologist, General Practitioner, counsellor, etc |
| Training given to intervention provider? | Yes, no |
| Level of education of intervention provider | BSc, MSc, PhD |
| Mode of mood management delivery | Individual, group |
| Location of mood management intervention | Hospital, participant's home, General Practitioner surgery, university, etc |
| Number of mood management sessions | Continuous variable |
| Length of mood management session (minutes) | Continuous variable |
| Was the mood management intervention tailored to participant? | Yes, no |
| Number of mood management sessions tailored to participant? | Continuous variable |
| Was participant adherence to mood management intervention measured? | Yes, no |
| Did participants adhere to the mood management intervention? | Yes, no or % |
| Was therapist adherence to mood management intervention measured? | Yes, no |
| Did therapists adhere to mood management programme? | Yes, no or % |

*The categories are likely to be further developed during data extraction to include new items.

► Interventions: the number of and function of behaviour change techniques used (ie, where sufficient details are not reported in text, we will attempt to obtain intervention protocols), and the presence or absence of TIDieR checklist items.

► Control: the number of and function of behaviour change techniques used (ie, where sufficient details are not reported in text, we will attempt to obtain intervention protocols), and the presence or absence of TIDieR checklist items.

► Measures of treatment effect, depression symptoms (study aim 3): for each trial arm, we will obtain mean depression scores and measure of variance at baseline and follow-up, mean differences and measures of variance from baseline to follow-up or differences in change between trial arms' scores from baseline to follow-up and measures of variance.

### Coding of TIDieR checklist
The 2013 Cochrane review[9] did not extract any information relevant to the TIDieR checklist[12]; these data are new to this review.

For study aim 1, we will use the TIDieR checklist[12] to determine if variations in mood management delivery impact on intervention effectiveness. We will use a modified version of TIDieR as not all items on the checklist are useful in the context of this study (table 1) (eg, 'Describe any rationale, theory or goal of the elements essential to the intervention'). Coding will be conducted separately by two researchers to confirm agreement.

### Coding of behaviour change intervention functions using the BCT
The 2013 Cochrane review[9] did not extract any information relevant to the BCT[13]; these data are new to this review.

For study aim 2, we will code the number of behaviour change techniques, categorise the behaviour change techniques according to their function and record whether the function was either absent or present during intervention delivery[13] (table 2). Coding will be conducted separately by two researchers to confirm agreement.

### Measures of treatment effect
► Smoking cessation (study aims 1 and 2): We will present treatment effects as RRs and 95% CIs.[14] RRs will be calculated as follows: (number of participants who quit smoking in the intervention group/number of participants randomised to intervention group) divided by (number of participants who quit smoking in control group/number of participants randomised to the control group).

**Table 2** Example of behaviour change functions and techniques[13]

| Behaviour change function | Examples of technique |
|---|---|
| Specific focus on behaviour and addressing motivation | Provide information on consequences of smoking and smoking cessation |
| | Boost motivation and self-efficacy |
| | Provide feedback on current behaviour |
| Specific focus on behaviour and maximising self-regulatory capacity/skills | Advise on changing routine |
| | Advise on environmental restructuring |
| | Set graded tasks |

► Difference in change in depression scores between trial arms (study aim 3): We will present the standardised mean difference (SMD), and 95% CIs of change in depression scores between treatment arms, from baseline to follow-up.

## Analysis

We will conduct analyses using Stata V.14 or Revman software, and use the following analytical procedures to address each study aim:

1. Do variations mood management delivery impact on intervention effectiveness? If there are sufficient data, we will conduct random effects metaregression models using the metareg command[15] in which modified TIDieR checklist items (see table 1) will be regressed on the study's effect estimate. First, univariate analyses will be conducted to determine the association between each item and the study effect size. Subsequently, items with the strongest association will be added to the metaregression model first, and all other variables will be added in turn regardless of significance in the univariate model.

2. Which behaviour change functions are most effective for smoking cessation in people with current depression? If there are sufficient data, we will conduct random effects meta-regression models using the metareg command[15] in which behaviour change functions (see table 2) will be regressed on the study's effect estimate. First, univariate analyses will be conducted to determine the association between each intervention function and the study effect size. Subsequently, variables with the strongest association will be added to the meta-regression model first, and all other variables will be added in turn regardless of significance in the univariate model.

3. What is the difference in change in depression scores between intervention and control arms? If there are sufficient data, we will use a generic inverse variance random effects model to pool the SMD of change in depression scores in treatment and control arms, from baseline to follow-up. We will use a random effects model as it incorporates heterogeneity both within and between studies.

Statistical heterogeneity: We will quantify statistical heterogeneity using $I^2$ which describes the percentage (%) of between-study variability due to heterogeneity rather than chance; values over 50% suggest substantial heterogeneity, and values over 75% suggest considerable heterogeneity.[14] $Tau^2$ will be used to test whether differences between studies' effect estimates are compatible with chance alone.[16]

Sensitivity and subgroup analyses: We will conduct sensitivity analyses to examine if the following study characteristics influence the meta-analysis results: study quality (as measured by Cochrane's Risk of Bias tool), loss-to-follow-up and severity of depression.

Assessment of publication bias: We will examine funnel plots for evidence of asymmetry and conduct egger tests for evidence of small study bias using the metabias command.[15]

## Ethics and dissemination

Ethical approval is not required for the conduct of this systematic review and meta-analysis. We will disseminate the findings of this work at international and national conferences, and to the UK Centre for Tobacco and Alcohol Studies Smokers' Panel.

## DISCUSSION

We will use the methods described in this protocol to determine: (1) if variations in delivery of mood management impact on smoking cessation intervention effectiveness in people with depression, (2) to examine which behaviour change functions are most effective for smoking cessation in people with depression and (3) examine the difference in change in depression scores between intervention and control arms.

We hold no strong hypotheses about which variations in mood management delivery/behaviour change functions will impact on treatment effectiveness. Potentially, intervention functions that focus on improving motivation to quit may strengthen the association between intervention and smoking cessation, as poor motivation is a hallmark symptom of depression. We do predict that at minimum smoking cessation intervention will not be associated with a worsening in depression, and that intervention may be associated with an improvement in depression scores when compared with control.[11]

## Clinical applications

If we can show that certain variations in delivery of mood management or behavioural support for smoking cessation are associated with higher abstinence rates, these data can be used by clinicians and researchers to optimise smoking cessation programmes for people with depression. Second, data pertaining to the impact of helping smokers with depression to quit smoking on depression symptoms will be imperative to smokers and clinicians.

**Author affiliations**

[1]Medical Research Council Integrative Epidemiology Unit, University of Bristol, Bristol, UK

[2]UK Centre for Tobacco and Alcohol Studies, School of Experimental Psychology, University of Bristol, Bristol, UK

[3]Nuffield Department of Primary Care Health Sciences, UK Centre for Tobacco and Alcohol Studies, University of Oxford, Oxford, UK

[4]Department of Epidemiology and Health Promotion, Public Health Service of Haaglanden (GGD Haaglanden), Hague, The Netherlands

[5]CAPHRI, Maastricht University, Maastricht, The Netherlands

[6]Jean Golding Institute for Data-Intensive Research, University of Bristol, Bristol, UK

[7]Centre for Academic Mental Health, Bristol Medical School, University of Bristol, Bristol, UK

**Contributors** All authors contributed to writing the manuscript and reviewed the final draft. GT, PA, RVM, DK, BS and MM all contributed towards study design. DT contributed towards writing the manuscript. GT and MM act as the guarantors of this review.

**Funding** GT is funded by Cancer Research UK Population Researcher Postdoctoral Fellowship award (reference: C56067/A21330). PA is supported by the NIHR Biomedical Research Centre and the NIHR CLAHRC. RVM reports no funding for this work. BS and DK are funded by The University of Bristol. MM would like to acknowledge funding from The MRC Integrative Epidemiology Unit at the University of Bristol which is supported by the Medical Research Council and the University of Bristol (MC_UU_12013/6). GT, MM and PA are members of the UK Centre for Tobacco and Alcohol Studies, a UKCRC Public Health Research: Centre of Excellence. Funding from the British Heart Foundation, Cancer Research UK, Economic and Social Research Council, Medical Research Council and the National Institute for Health Research, under the auspices of the UK Clinical Research Collaboration, is gratefully acknowledged.

**Disclaimer** The funders have had no role in developing the protocol or study design.

**Competing interests** All authors completed ICMJE form for disclosure of potential conflicts of interest. GT, DT, RVM, DK and BS report no competing interests. PA reports non-financial support from GSK, outside the submitted work. MM reports grants from Pfizer, outside the submitted work.

**Provenance and peer review** Not commissioned; externally peer reviewed.

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
