## [Reviewer comments · BMJ Open]

ARTICLE DETAILS

TITLE (PROVISIONAL)	Impact of variation in intervention delivery and intervention functions on the effectiveness of behavioural and mood management interventions for smoking cessation in people with depression: A systematic review and meta-analysis protocol.
AUTHORS	Taylor, Gemma; Aveyard, Paul; Van der Meer, Regina; Toze, Daniel; Stuijzand, Bobby; Kessler, David; Munafo, Marcus

VERSION 1 – REVIEW

REVIEWER	Vanessa Clark University of Newcastle, Australia
REVIEW RETURNED	08-Aug-2017

GENERAL COMMENTS	This is an interesting topic with well defined research questions. The inclusion of the search criteria would be useful, so that it can be evaluated as part of the protocol. It will make an interesting paper, that I look forward to reading. Statistics were unable to be evaluated, as they have not been completed yet (as it is a protocol), but the plan is adequate.
--

REVIEWER	Emily Stockings National Drug and Alcohol Research Centre (NDARC), UNSW Sydney, Australia
REVIEW RETURNED	08-Aug-2017

GENERAL COMMENTS	This manuscript outlines the proposed methods for a systematic review and meta-analysis to further explore the impact of variations in intervention delivery and content on smoking cessation outcomes among smokers with current depression. The review builds on an existing Cochrane review. The topic is one of importance for researchers in the field of smoking cessation given the known high rates of smoking and overall weak cessation outcomes in this group. Overall the described methods are sound. I have some points that require further clarification below: 1. The abstract states the review has been registered on PROSPERO, but the methods (p5, line 5) state that the review “will be” – the registration status should be consistent throughout. 2. A clearer distinction needs to be made between the original Cochrane Review methods, and any alterations to these for the current review. I suggest moving the “Search strategy” component to the start of the methods section in order to establish that the current review is based on the review by Van der Meer 2013.
---

	Then, authors should clearly outline what components of this protocol are based on Cochrane methods (e.g. inclusion criteria), and what is being added specifically for this review (e.g. the outcome of change in depression scores). 3. Are any of the authors on this paper also authors of the Cochrane review? If so this should be stated, and/or methods for collaborating with the Cochrane authors (e.g. data sharing) should be outlined. Given the high reliance on the existing review and access to unpublished review updates, feasibility needs to be demonstrated. 4. Page 5, line 34 – state the intended year/month that the updated review will be conducted. If month is unknown, simply add the year. Will this update be for the original Cochrane review, or for this specific review only? 5. Data extraction (pages 5-6) – I assume most of this has already been completed by the original review authors. If so, does the new review team have access to these data? What processes will occur to ensure data sharing. Which of these variables (if any) have not previously been extracted and are new to this review? This needs delineating. 6. I would have expected a brief discussion section summarising what this additional review will add to the original Cochrane review, what results are hypothesised or anticipated, if previous research has been conducted in this area, what is known, what needs to be known, and what the clinical and/or research applications are. 7. Author contributions includes work that has not occurred yet, namely “DT is contributing to data extraction”. Any work that has not yet commenced, or that did not directly relate to this manuscript as it stands should be removed from this section.
--	--

VERSION 1 – AUTHOR RESPONSE

Reviewer: 1

Reviewer Name: Vanessa Clark

Institution and Country: University of Newcastle, Australia

Please state any competing interests: None declared

Reviewer 1 comments: “This is an interesting topic with well-defined research questions. The inclusion of the search criteria would be useful, so that it can be evaluated as part of the protocol. It will make an interesting paper, that I look forward to reading. Statistics were unable to be evaluated, as they have not been completed yet (as it is a protocol), but the plan is adequate.”

Authors’ reply: Thank you for your interest in our review. The search strategy and criteria have been updated with further details on page 5, lines 129 to 140, as follows:

“We will include relevant studies identified by a previously conducted Cochrane review of smoking cessation interventions for people with depression, and from the Cochrane review update due to commence this year.(van der Meer, Willemsen, Smit, & Cuijpers, 2013) Studies have been identified from Cochrane Central Register of Controlled trials (CENTRAL), MEDLINE, EMBASE, and PsycINFO using search terms related to ‘depression’, and ‘tobacco’ or ‘smoking’ as recommended by the Tobacco Addiction Group and the Cochrane Depression, Anxiety and Neurosis Group. See the Tobacco Addiction Group Module in The Cochrane Library for full search strategies and the list of other resources searched (van der Meer et al., 2013). This search strategy will be updated for additional relevant studies published from 2013.”

Reviewer: 2

Reviewer Name: Emily Stockings

Institution and Country: National Drug and Alcohol Research Centre (NDARC), UNSW Sydney, Australia

Please state any competing interests: None declared

Reviewer 2 comments: This manuscript outlines the proposed methods for a systematic review and meta-analysis to further explore the impact of variations intervention delivery and content on smoking cessation outcomes among smokers with current depression. The review builds on an existing Cochrane review. The topic is one of importance for researchers in the field of smoking cessation given the known high rates of smoking and overall weak cessation outcomes in this group. Overall the described methods are sound. I have some points that require further clarification below:

1. The abstract states the review has been registered on PROSPERO, but the methods (p5, line 5) state that the review “will be” – the registration status should be consistent throughout.

Authors’ reply: Thank you for highlighting this inconsistency. We’ve updated this throughout the manuscript.

Reviewer 2 comments: A clearer distinction needs to be made between the original Cochrane Review methods, and any alterations to these for the current review. I suggest moving the “Search strategy” component to the start of the methods section in order to establish that the current review is based on the review by Van der Meer 2013. Then, authors should clearly outline what components of this protocol are based on Cochrane methods (e.g. inclusion criteria), and what is being added specifically for this review (e.g. the outcome of change in depression scores).

Authors’ reply: We have moved the “search strategy” section to the beginning of the methods section, and outlined which components of this protocol are based on Cochrane methods, and what is being added by this review. Please see page 5, lines 128-140.

Reviewer 2 comments: Are any of the authors on this paper also authors of the Cochrane review? If so this should be stated, and/or methods for collaborating with the Cochrane authors (e.g. data sharing) should be outlined. Given the high reliance on the existing review and access to unpublished review updates, feasibility needs to be demonstrated.

Authors’ reply: We have added the following description of our methods of collaborating with the Cochrane review lead author to the “Search strategy” section.

“This search strategy will be updated for additional relevant studies published from 2013. RV is the lead author for the Cochrane review published in 2013, and will lead on the Cochrane update of this review. To avoid duplicating efforts across teams and given the high reliance of Cochrane methods, RV will share the eligible studies prior to data extraction of the Cochrane update. We predict that this will take place early 2018.”

Reviewer 2 comments: Page 5, line 34 – state the intended year/month that the updated review will be conducted. If month is unknown, simply add the year. Will this update be for the original Cochrane review, or for this specific review only?

Authors' reply: We have added this information to the search strategy section. See page 5, lines 139-140.

Reviewer 2 comments: Data extraction (pages 5-6) – I assume most of this has already been completed by the original review authors. If so, does the new review team have access to these data? What processes will occur to ensure data sharing. Which of these variables (if any) have not previously been extracted and are new to this review? This needs delineating.

Authors' reply: We have re-structured the methods section to clarify which variables have, and have not previously been extracted, and which variables are new to this review. Additionally, we have clarified methods for sharing data between the reviews on page 5, lines 137-139.

Reviewer 2 comments: I would have expected a brief discussion section summarising what this additional review will add to the original Cochrane review, what results are hypothesised or anticipated, if previous research has been conducted in this area, what is known, what needs to be known, and what the clinical and/or research applications are.

Authors' reply: Many thanks for suggesting this. We've added the following to the discussion section to page 10, lines 256-268:

“Discussion

We will use the methods described in this protocol to determine: 1) if variations in delivery of mood management impact on smoking cessation intervention effectiveness in people with depression, 2) to examine which behaviour change functions are most effective for smoking cessation in people with depression, and 3) examine the difference in change in depression scores between intervention and control arms.

We hold no strong hypotheses about which variations in mood management delivery/behaviour change functions will impact on treatment effectiveness. Potentially, intervention functions that focus on improving motivation to quit may strengthen the association between intervention and smoking cessation, as poor motivation is a hallmark symptom of depression. We do predict that at minimum smoking cessation Intervention will not be associated with a worsening in depression, and that intervention may be associated with an improvement in depression scores when compared to control (Taylor et al., 2014).

Clinical applications

If we are able to show that certain variations in delivery of mood management or behavioural support for smoking cessation are associated with higher abstinence rates, these data can be used by clinicians and researchers to optimise smoking cessation programmes for people with depression. Second, data pertaining to the impact of helping smokers with depression to quit smoking on depression symptoms will be imperative to smokers and clinicians.”

Reviewer 2 comments: Author contributions includes work that has not occurred yet, namely “DT is contributing to data extraction”. Any work that has not yet commenced, or that did not directly relate to this manuscript as it stands should be removed from this section.

Authors' reply: Thank you, we've removed that sentence.

Reference

Taylor, G., McNeill, A., Girling, A., Farley, A., Lindson, N., & Aveyard, P. (2014). Change in mental health after smoking cessation: systematic review and meta-analysis. *BMJ*, 348.

van der Meer, R. M., Willemsen, M. C., Smit, F., & Cuijpers, P. (2013). Smoking cessation interventions for smokers with current or past depression. *The Cochrane Library*. JOUR.

VERSION 2 – REVIEW

REVIEWER	Vanessa Clark University of Newcastle, Australia
REVIEW RETURNED	13-Sep-2017

GENERAL COMMENTS	An interesting protocol and I wish the authors well in their review.
--

REVIEWER	Emily Stockings National Drug and Alcohol Research Centre, UNSW Sydney
REVIEW RETURNED	20-Sep-2017

GENERAL COMMENTS	The authors have comprehensively addressed my comments. The distinction between the original Cochrane Review and the current review is very clear, and this protocol is now a good example of how effective collaboration with Cochrane authors can bolster research output and allow investigation of more specific research questions. I have no further comments and look forward to reading the final review.
---

VERSION 2 – AUTHOR RESPONSE

We would like to thank our reviewers and the editor for their comments. We have made the editors suggestions, and have also proof read the manuscript again. We attach a tracked changes and a clean version of the manuscript.